# Phlebotomine Sand Flies in Southern Thailand: Entomological Survey, Identification of Blood Meals and Molecular Detection of *Trypanosoma* spp.

**DOI:** 10.3390/insects13020197

**Published:** 2022-02-14

**Authors:** Jirayu Buatong, Vit Dvorak, Arunrat Thepparat, Kanaphot Thongkhao, Surachart Koyadun, Padet Siriyasatien, Theerakamol Pengsakul

**Affiliations:** 1Faculty of Medical Technology, Prince of Songkla University, Songkhla 90110, Thailand; jirayu.mook@gmail.com; 2Department of Parasitology, Faculty of Science, Charles University, 12844 Prague, Czech Republic; icejumper@seznam.cz; 3Department of Agricultural Technology, Faculty of Science, Ramkhamhaeng University, Hua Mak, Bang Kapi, Bangkok 10240, Thailand; arunratthepparat@gmail.com; 4Office of Disease Prevention and Control, Region 11, Nakhon Si Thammarat 80000, Thailand; kanaphot.t@gmail.com (K.T.); thvbdosk@yahoo.com (S.K.); 5Vector Biology and Vector Borne Disease Research Unit, Department of Parasitology, Faculty of Medicine, Chulalongkorn University, Bangkok 10330, Thailand; padet.s@chula.ac.th

**Keywords:** sand flies, *Sergentomyia*, *Phlebotomus*, *Trypanosoma* sp., blood meal source

## Abstract

**Simple Summary:**

Phlebotomine sand flies (Diptera: Psychodidae) are hematophagous insects, and many species serve as vectors of various human and animal pathogens, including *Leishmania* and *Trypanosoma* protozoa. In Thailand, the first case of autochthonous leishmaniasis was reported 62 years ago. At present, the number of human cases is increasing in different regions of the country, but most cases are reported from southern Thailand. Therefore, we studied the potential transmission of *Leishmania* and *Trypanosoma* by sand flies in three provinces of southern Thailand, and analyzed blood sources of engorged sand fly females. We detected *Trypanosoma* sp. DNA in *Sergentomyia barraudi*, *S. indica*, *S. khawi* and *Idiophlebotomus asperulus* but no *Leishmania* spp. DNA. Moreover, bloodmeal analysis revealed that *Trypanopsoma*-positive females of *S. barraudi* and *Sergentomyia* sp. fed on dogs and humans, respectively. The results of this study contribute to the knowledge of leishmaniasis and trypanosomiasis presence and sand fly feeding behavior in southern Thailand.

**Abstract:**

An entomological survey at rural and cavernicolous localities in four provinces in southern Thailand provided 155 blood-fed females of sand flies (Diptera: Psychodidae) that were identified based on morphological characters as *Idiophlebotomus asperulus* (*n* = 19), *Phlebotomus stantoni* (*n* = 4), *P. argentipes* (*n* = 3), *Sergentomyia anodontis* (*n* = 20), *S. barraudi* (*n* = 9), *S. hamidi* (*n* = 23), *S. hodgsoni* (*n* = 4), *S. hodgsoni hodgsoni* (*n* = 32), *S. indica* (*n* = 5), *S. iyengari* (*n* = 2), *S. khawi* (*n* = 17), *S. silvatica* (*n* = 11) and *Sergentomyia* sp. (*n* = 6). The dominant species in this study was *S. hodgsoni hodgsoni*, which was collected specifically in a Buddha cave. Screening for DNA of parasitic protozoans revealed eight specimens (5.16%) of four species (*S. barraudi*, *S. indica*, *S. khawi* and *Id. asperulus*) positive for *Trypanosoma* sp., while no *Leishmania* spp. DNA was detected. Blood meals of engorged females were identified by PCR-Restriction Fragment Length Polymorphism (PCR-RFLP) assay on a fragment of cytochrome *b* (*cyt b*) gene with a success rate 36%, humans, dogs, and rats being determined as sources of blood. Bloodmeal analysis of two *Trypanopsoma*-positive females (*S. barraudi* and *Sergentomyia* sp.) identified blood from dogs and humans, respectively. Our findings indicate that *S. barraudi*, *S. indica*, *S. khawi* and *Id. asperulus* may be incriminated in circulation of detected *Trypanosoma* spp.

## 1. Introduction

Phlebotomine sand flies (Diptera: Psychodidae) are hematophagous insects, and many species serve as vectors of *Leishmania* spp. [1,2] to humans and animals resulting in Leishmaniasis. The DNA of *Trypanosoma* spp. has also been detected in sand flies [2,3,4]; however, sand flies have not been incriminated as a *Trypanosoma* vector. Leishmaniasis is a vector-borne disease that continues to be a major health problem in many regions of the world [5]. In Thailand, four human-infecting *Leishmania* were reported [6]; *Leishmania donovani* and *L. infantum* belong to the long-time known subgenus *Leishmania*, *L. martiniquensis* and *L. orientalis* (originally described as *L. siamensis*) are classified within a newly established subgenus *Mundinia* [7]. Human diseases caused by these four species show various clinical symptoms, manifested as visceral leishmaniasis (VL), caused by *L. donovani*, *L. infantum*, *L. martiniquensis* and *L. orientalis* [8,9,10,11]; cutaneous leishmaniasis (CL), caused by *L. donovani*, *L. infantum*, *L. martiniquensis* and *L. orientalis* [7,12,13,14]; and diffuse cutaneous leishmaniasis (DCL) caused by *L. infantum*, *L. martiniquensis* and *L. orientalis* [10,15,16]. Since the first imported case in 1960, autochthonous human leishmaniasis has increased to twenty cases [6]. Most of these local human cases were reported in southern Thailand in Phang-Nga, Satun, Krabi, Trang, Nakhon Si Thammarat, Songkhla, Phattalung, and Surat Thani provinces, highlighting the importance to investigate leishmaniasis prevalence in these regions. Two previous studies detected DNA of *L. martiniquensis* in *S. barraudi*, *S. gammea* and *S. iyengari* from endemic leishmania area in Trang province, southern Thailand [17,18]. Recently, the novel species *L. orientalis* was detected from a cutaneous leishmaniasis patient in northern Thailand [7], and *L. orientalis* has been detected in field-caught *S. iyengari* in southern Thailand. Although human trypanosomiases caused by species of the genus *Trypanosoma* have never been reported in Thailand, *Trypanosoma evansi* and *Trypanosoma lewisi*, two species that can be transmitted from animals to humans, and were incriminated in rare human infections [19,20,21,22], were detected in animals [23,24]. Recently, *Trypanosoma* sp. DNA was detected in *S. khawi* and *P. stantoni* in southern Thailand [2,25]. These findings suggest that sand flies in southern Thailand may be incriminated in transmission of both *Leishmania* and *Trypanosoma* protozoans. Although *Trypanosoma* DNA detection in sand fly species has been reported continuously, the complete information of *Trypanosoma* life cycle remains uncertain, while the life cycle of *Leishmania* in sand flies has been suggested as a natural vector of *Leishmania* [26,27]. However, little is known about the biology and feeding preferences of sand flies that may provide important insight into their transmission cycle and potential reservoir hosts.

Blood meal analyses provide important insight into the dynamics and transmission route of sand fly-borne diseases. The immunological techniques such as enzyme-linked immunosorbent assay (ELISA) were initially deployed to identify sand fly bloodmeals [28,29], later replaced by molecular tools with higher accuracy. The amplification of the *cyt b* gene using polymerase chain reaction (PCR) technique and subsequent sequencing is currently widely used for blood meal analysis. However, it remains less cost- and time-effective for analysis of field samples in larger field surveys [30]. The PCR-Restriction Fragment Length Polymorphism (PCR-RFLP) of the *cyt b* gene decreases the cost of the analysis as it does not require the sequencing step and it has been successfully used to identify sand fly blood meals in many endemic regions, including Brazil [31,32] and Spain [33]. However, it has not yet been applied to sand flies in Thailand. The comparison of potential of PCR-RFLP of the *cyt b* gene and amplification of the *cyt b* gene using the PCR technique and subsequent sequencing has been studied [33], showing that the PCR-RFLP technique is an efficient and reliable alternative equal to PCR and subsequent sequencing, with lower cost and quicker results. The PCR-RFLP technique is based on amplifying the *cyt b* gene of the sandflies blood meal, then cleaving by restriction endonuclease enzymes on recognition sites and cleaving on DNA fragments. The cleaved DNA fragment pattern of the *cyt b* gene could be shown distinguishing between vertebrate species [33]. However, the PCR-RFLP technique necessitates the use of two or more restriction enzymes in order to obtain conclusive results [33]. Therefore, this study aimed to identify sources of blood in engorged sand fly females to understand their host preferences. In addition, the specimens were also screened for the presence of *Leishmania* and *Trypanosoma* DNA.

## 2. Materials and Methods

### 2.1. Ethics Statement

The method for sand fly collecting and specimen preparation in this study was certified by the Institutional Animal Care and Use Committee, Prince of Songkla University, under reference number 2561-10-021.

### 2.2. Sand Fly Collection

Insects were collected at five localities in four provinces of southern Thailand: from four rural settlements in Phang-Nga (site PNA), Satun (site STN1), Songkhla (site SKA), Surat Thani (site SNI) provinces, and one Buddha cave in Satun province (site STN2) (Figure 1), using Center for Disease Control (CDC) light traps (John W. Hock Co., Gainesville, FL, USA) placed 5 to 10 cm above the ground. Ten traps were deployed at each site monthly in the early rainy season in June and August 2015, and in the late rainy season in January 2015, from sunset to sunrise (6.00 p.m.–6.00 a.m.). Insects were sacrificed with ethyl acetate in a plastic bag. Sand flies were collected from the total catch insects on Petri dishes under a stereomicroscope (Olympus, Tokyo, Japan) and preserved in a 70% ethanol solution until processing.

### 2.3. Sample Preparation

Blood-fed sand fly females were selected and placed individually in a 1.5 mL microtube containing 70% ethanol. The head of each individual sand fly was dissected on a sterilized glass slide and mounted into Hoyer’s medium [34]. The individual thorax and abdomen were transferred to a sterile 1.5 mL microtube with 70% ethanol and stored at −20 °C for later DNA extraction.

### 2.4. Morphological Identification of Sand Flies

The species of sand flies was identified using taxonomic keys [35,36,37,38] based on their species-specific morphological characters of cibarium (shape and color of pigment patch, number of vertical and horizontal teeth) and pharynx under light microscope (Olympus, Tokyo, Japan).

### 2.5. Genomic DNA Extraction

The thorax and abdomen of each sand fly stored in 70% ethanol were transferred into a new 1.5 mL microtube, dried at room temperature to remove the remnants of ethanol and homogenized using a disposable polypropylene pestle. The genomic DNA was extracted using the DNeasy Blood and Tissue Kit (Qiagen, Hilden, Germany) according to the manufacturer’s protocol, with a final elution by 200 µL of elution buffer, and then stored at −20 °C.

### 2.6. Polymerase Chain Reaction (PCR) Analysis and DNA Sequencing

The concentration of genomic DNA was determined using a NanoDrop 2000c spectrophotometer (Thermo Scientific, Waltham, MA, USA). The cytochrome c oxidase subunit I (COI) gene was amplified using specific primers LepF (5′-ATTCAACCAATCATAAAGATATTGG-3′) and LepR (5′-AAACTTCTGGATGTCCAAAAAATCA-3′) to obtain the PCR products approximately 658 bp [39] to identify *Trypanosoma* DNA positive sand flies and confirm the species level. The PCR reaction was performed in a total volume of 25 µL, containing 10.35 µL nuclease-free water, 0.5 µL of each primer (10 µM), 7.65 µL of My Taq™ HS Red Mix (BioLine, Taunton, MA, USA) and 6 µL of DNA template. The PCR reactions were performed in Eppendorf Mastercycler ^®^ Nexus thermal cycler (Eppendorf, Hamburg, Germany) following the published thermal profile as described previously [39]. The PCR products were stained with MaestroSafe TM Nucleic acid loading dye (Maestrogen, Hsinchu, Taiwan), loaded onto a 1.5% agarose gel and visualized under UV light.

PCR amplification of the internal transcribed spacer 1 (ITS1) region of *Leishmania* was done using primers LeF (5′-TCCGCCCGAAAGTTCACCGATA-3′) and LeR (5′-CCAAGTCATCCATCGCGACACG-3′) [40]. However, this primer set is also complementary to other Trypanosomatidae sequences [40]. Therefore, the amplification of the ITS1 region of *Trypanosoma* was also performed using this primer set. The total volume of PCR reaction was 25 µL and consisted of 11.2 µL of nuclease-free water, 0.4 of each primer (10 µM), 7 µL of My Taq™ HS Red Mix (BioLine, USA) and 6 µL of DNA template. The PCR was performed following the profile as previously described [40].

The PCR products were purified using TIANquick Midi Purification Kit (TIANGEN BIOTEC, Beijing, China). The purified PCR products were sent to a sequencing service (Macrogen, Seoul, Korea) for direct sequencing using the same primers as those in the PCR reactions.

### 2.7. Identification of Blood Meal Source by PCR-RFLP

The *cyt b* gene of mitochondrial DNA was amplified from the genomic DNA. The specific primers of vertebrates were used: BM1 (5′-CCCCTCAGAATGATATTTGTCCTCA-3′) and BM2 (5′-CCATCCAACATCTCAGCATGATGAAA-3′) [31]. The total volume of PCR was 25 µL, containing 11 µL of nuclease-free water, 7 µL of My Taq™ HS Red Mix (BioLine, Taunton, MA, USA), 0.5 of each primer (10 µM) and 6 µL of DNA template. The reaction was performed in a thermal cycler (Eppendorf, Hamburg, Germany) following the PCR profile [41]. The PCR products sized 358 bp were obtained and used for blood meal source analysis by restriction fragment length polymorphism (RFLP) technique. The PCR products were cut at the specific site by restriction enzymes *Hae* III, *Rsa* I and *Aci* I (Thermo Fisher scientific, Inc., Waltham, MA, USA). PCR product digestion was performed in a total volume of 30 µL containing 17 µL of nuclease free water, 10 µL of PCR product, 2 µL of 10X FastDigest Green Buffer, 1 µL of FastDigest enzyme and 10 µL of PCR product, and then incubated at 37 °C for 5 min. The digested PCR product was loaded onto 1.5% agarose gel for fragment size evaluation alongside a molecular weight marker (BioLine, Taunton, MA, USA). The expected fragments of *cyt b* PCR products digested by endonuclease enzymes *Hae* III, *Rsa* I and *Aci* I [31] are shown in Table 1 and used for blood meal source identification. The patterns of digested DNA fragments of the three endonuclease enzymes in each blood meal source show distinct differences.

### 2.8. Phylogenetic Analysis of Sand Fly Species and Detected Parasites

The DNA sequences of sand flies and *Trypanosoma* were checked for ambiguous base calls and assembled by BioEdit version 7.2.5 [42]. The assembly sequences were compared to the GenBank database using BLASTn tool (https://blast.ncbi.nlm.nih.gov/Blast.cgi accessed on 9 November 2021). Phylogenetic trees were constructed by the maximum likelihood (ML) method using Molecular Evolutionary Genetics Analysis version 6.0 (MEGA 6.0) software [43].

## 3. Results

### 3.1. Collection and Morphological Identification of Sand Flies

A total of 384 sand flies were collected from localities in Phang-Nga, Satun, Songkhla and Surat Thani provinces. Of these, 155 blood-fed females were selected for *Leishmania* spp. and *Trypanosoma* spp. detection and blood meal source analysis. The highest number of blood-fed females was collected from Satun in a Buddha cave (69 individuals, 44.5%) and in the vicinity of rural settlements (46 individuals, 29.68%), followed by collections near rural settlements from Songkhla, Phang-Nga, and Surat Thani, with 24 (15.48%), 11 (7.10%) and 5 (3.23%) individuals, respectively. These sand flies were classified in the genera *Idiophlebotomus*, *Phlebotomus* and *Sergentomyia*. Based on morphological characteristics, twelve species were identified: *Idiophlebotomus asperulus* (*n* = 19), *P. stantoni* (*n* = 4), *P. argentipes* (*n* = 3), *S. anodontis* (*n* = 20), *S. barraudi* (*n* = 9), *S. hamidi* (*n* = 23), *S. hodgsoni* (*n* = 4), *S. hodgsoni hodgsoni* (*n* = 32), *S. indica* (*n* = 5), *S. iyengari* (*n* = 2), *S. khawi* (*n* = 17), *S. silvatica* (*n* = 11) and *Sergentomyia* sp. (*n* = 6). The dominant species in this study was *S. hodgsoni hodgsoni*, which was found only in one cave in Satun province (Table 2).

### 3.2. Detection of Leishmania and Trypanosoma spp. in Sand Flies

Screening of the genomic DNA of all 155 engorged females for *Leishmania* DNA by PCR amplification of ITS1 region provided eight positive samples (5.16%). Four of them were detected in samples from Satun province (ST43, ST134, ST143 and ST139), three from Songkhla province (SK97, SK109 and SK91) and one (PN71) from Pang-Nga province. ITS1 sequences obtained were compared with sequences from GenBank by nucleotide BLAST (blastn) search analysis, showing unexpectedly best match with *Trypanosoma* spp. rather than *Leishmania* spp. Phylogenetic analysis of ITS1 sequences by the maximum likelihood (ML) method showed that sequence LSK97 group together with *Trypanosoma congolense*, *T. rangeli* and *T. lewisi*, while the sequences LSK91 and LPN71 were closely related to *Trypanosoma minasense* (AB362411), and the sequence LST143 is closely related to *Trypanosoma avium* (AY929322) in Clade A. The samples LSK91, LSK97, LPN71 and LST143 showed low sequence identity (<50%) with their closest strains; therefore, the sequences LSK91, LSK97, LPN71 and LST143 were identified as *Trypanosoma* sp. based on ITS1 region (Figure 2). When considering the sequences in Clade B, samples LSK109, LST43, LST134 and LST139 group together with *Trypanosoma* sp. (KJ467211) with 62% bootstrap (Figure 1). The sequence LST43 showed 86.8% sequence similarity to *Trypanosoma* sp. (KJ467211), while other samples in Clade B showed very low sequence similarity to *Trypanosoma* sp. (KJ467211) in the range of 48–55%. Therefore, the sequences LSK109, LST43, LST134 and LST139 were identified as *Trypanosoma* sp. The ITS1 sequences of *Trypanosoma* sp. in this study were deposited in the GenBank database under the accession number OL332783-OL332790.

### 3.3. Molecular Identification of Trypanosoma-Positive Sand Flies

Species identity of eight *Trypanosoma*-positive sand flies was confirmed using PCR amplification and sequencing analysis of COI gene. The phylogenetic tree was constructed based on COI gene sequences of sand flies and related sand fly sequences derived from the GenBank and BOLD databases using the ML method with 1000 replicates of bootstrapping. Based on the sequences obtained, four species with high sequence similarity (99.8–100%) were identified: *S. barraudi* (ST143), *S. indica* (SK91), *S. khawi* (SK97, SK109) and *Id. asperulus* (ST134, PN71). Two samples (ST43 and ST139) were identified as *Sergentomyia* sp. with no species designation due to lack of reference sequences (Figure 3). These two sand flies were suspected to be novel species. The partial COI gene sequences of sand fly sequences were deposited in GenBank with the accession number OK576207-OK576214.

### 3.4. Identification of Blood Source by RFLP

The genomic DNA of all 155 blood-fed females was used as a template for the amplification of the *cyt b* gene followed by RFLP assay. The 358 bp PCR product was successfully amplified (Figure 4a) for 56 samples (36.13%): 50 of *Sergentomyia* spp., 3 of *Id. asperulus* and 3 of *P. argentipes*. RFLP patterns revealed that human was the main blood meal source, as it was detected in 33 samples, followed by rat and dog blood identified in 14 and 9 samples, respectively (Table 2). Briefly, the restriction pattern for *cyt b* PCR products of human blood samples by *Hae* III endonuclease produced bands of 233 and 124 bp, while in the case of PCR products of dog and rat blood samples, no cleavage site was found. The pattern of rat DNA digested by *Rsa* I generated fragments with sizes of 267, 59, 31 bp, whereas the restriction site on *cyt b* PCR product from human and dog samples was not found. The fragment pattern of human DNA digested by *Aci* I resulted in fragments of 189, 113, 55 bp. Dog and rat DNA samples were not digested by *Aci* I (Table 1). No mixed blood meals were identified. Human was the most often detected host for *S. hamidi* and *S. silvatica*. The fragment of *cyt b* gene could not be amplified from blood-fed *P. stantoni*, *S. indica* and *S. iyengari*. Of eight *Trypanosoma*-positive sand flies, blood source of only two samples (ST43 and ST143) was successfully determined (Figure 4b), while for six remaining samples (SK91, SK97, SK109, ST134, ST139 and PN71) amplification of the *cyt b* gene fragment failed. Based on the RFLP patterns of samples ST43 and ST143, the source of blood was determined as human and dog, respectively.

## 4. Discussion

In this study, twelve sand fly species, namely *Idiophlebotomus asperulus*, *P. stantoni*, *P. argentipes*, *S. anodontis*, *S. barraudi*, *S. hamidi*, *S. hodgsoni*, *S. hodgsoni hodgsoni*, *S. indica*, *S. iyengari*, *S. khawi*, *S. silvatica* and *Sergentomyia* sp. were collected from Phang-Nga, Satun, Songkhla and Surat Thani provinces, which are regions so far endemic for human leishmaniasis. All eleven sand fly species except *S. hodgsoni hodgsoni* were found in rural settlements. In contrast, *S. hodgsoni hodgsoni* was the most abundant species specifically found in a Buddha cave in Satun province, while *S. khawi* was not found in any cave. This finding agrees with earlier work suggesting that *S. hodgsoni hodgsoni* is the most common species collected from caves in Kanchanaburi province, Thailand [44], so we confirm that it is indeed a cavernicolous species. The rest of the identified sand fly species were collected from both rural settlements and caves. The presence of 12 species at the studied localities comprises nearly half of all sand fly species reported in Thailand so far. In general, sand fly fauna in Thailand is still not fully investigated and rather poorly documented. The last published checklist includes 26 species of four genera [44], but later records and taxonomical reassessments added several more species such as *S. hivernus* [25] or *S. raynali* [45]. We may expect that future research that combines complementary morphological and molecular approaches will provide further insight into the true local diversity, and potentially may lead to recognition and taxonomical description of new species of sand flies in Thailand. As some species seem to be cave-dwellers, special attention will be given to vast cavernicolous habitats which remain largely unexplored and may reveal a score of so far undescribed species.

Sequencing analysis of vertebrate mitochondrial cytochrome *b* gene has been widely used to identify blood sources of various hematophagous insects, including sand flies [46]. In our study, we deployed this approach to successfully identify blood meals in approximately one-third of engorged females collected. This success rate reflects the fact that amplification of the host gene is difficult in later stages of blood meal digestion due to the negative effect of progressing digestive enzymes. This shortcoming could be overcome in future studies by the deployment of protein-based MALDI-TOF peptide mapping that has recently been demonstrated to also allow successful blood identification in blood meals with more advanced blood digestion [47]. In our study, humans were the major blood meal source of sand flies, followed by rats and dogs. No mixed blood meals were detected. Several species demonstrated opportunistic feeding behavior: blood of two or even three hosts were detected in different females of *P. argentipes*, *S. anodontis*, *S. barraudi*, *S. silvatica* and *S. hodgsoni hodgsoni*. This *S. hodgsoni hodgsoni* which, despite its exclusive occurrence in a cavernicolous habitat, was demonstrated to feed on dogs, humans and rats alike. As sand flies are generally regarded as rather poor fliers with limited range of host seeking, we may assume that they readily bite occasional human visitors of the caves. Moreover, humans were the most preferred hosts of *S. silvatica* and *S. hamidi*, although the low number of analyzed females does not allow credible assumptions about the anthropophily of these species. Blood meal analysis of females identified as *S. khawi*, *S. indica* and *Id. asperulus* that were shown to harbor DNA of *Trypanosoma* sp. unfortunately failed to provide identification of their blood meals, thus not providing any insight regarding the potential involvement of these three species in the transmission of these protozoans.

Hematophagous sand fly females serve as vectors of different pathogens, most importantly protozoans of the genus *Leishmania* that cause disease with varying clinical outcomes. Albeit Thailand is traditionally not regarded as a country heavily burdened by human leishmaniases, numbers of autochthonous cases increased in the last six decades, and four *Leishmania* species were recorded as their causative agents. So far, none of the sand fly species naturally occurring in the country were conclusively proven as a vector; however, various sand flies of the genera *Sergentomyia* spp. and *Phlebotomus* spp. are suspected vectors, since *Leishmania* DNA was detected by PCR assays within the field-collected specimens of these species: DNA of *L. martiniquensis* has been detected in *S. khawi* [2], and DNA of *L. siamensis* (later synonymized with *L. martiniquensis*) in *S. gemmea, S. barraudi* and *S. iyengari* [17,18,48]. In this study, we did not detect *Leishmania* DNA in any of 155 analyzed specimens. This correlates with a previous study from Satun province, which did not detect any *Leishmania* DNA in field-collected specimens of *P. stantoni*, *P. argentipes*, *S. gemmea*, *S. indica*, *S. barraudi*, *S. iyengari*, *S. bailyi*, *S. perturbans* and *S. silvatica* [49].

Besides *Leishmania*, which is the most important sand fly-borne parasitic protozoan, sporadic infections of *Trypanosoma* spp. were previously reported in *Phlebotomus* spp. [50] and *Sergentomyia* spp. [51] from various geographical regions. In Thailand, *Trypanosoma* sp. was recently detected in a single specimen of *P. stantoni* [25], while another study detected *T. noyesi* in *S. anodontis* and *P. asperulus* and another yet undescribed *Trypanosoma* sp. in these two sand fly species, along with *P. betisi* collected in endemic and nonendemic leishmaniasis areas in southern Thailand [2]. The results from our study correlate to those detected *Trypanosoma* sp. infected *Id. asperulus*, and may indicate the potential transmitter of *Trypanosoma* spp. in Thailand, as recently reported in terms of an infected *Trypanosoma* sp. found in *S. barraudi*, *S. indica* and *S. khawi*. *Trypanosoma* sp. LST43 detected in *Sergentomyia* sp. from Satun province in the current study belongs to *Trypanosoma* sp. (KJ467211) sequence from GenBank database with high sequence identity. *Trypanosoma* sp. (KJ467211) DNA was detected in *P. stantoni* from Songkhla province, southern Thailand [25]. From this result, we conclude that two sequences of *Trypanosoma* sp. LST43 may be the same species as *Trypanosoma* sp. (KJ467211). Although to date no human cases of trypanosomiasis were reported in Thailand so far, trypanosomiasis caused by *Trypanosoma evansi* and *Trypanosoma lewisi* were reported in animals [52,53]. Thus, precise species identification of sand fly-borne *Trypanosoma* spp. and further assessment of the potential role of different sand fly species as their vectors remain important factors to consider for adequate prevention planning. The specific primer for species-level identification of *Trypanosoma* spp. will be designed in our future work.

## 5. Conclusions

This study reports for the first time a detection of *Trypanosoma* sp. DNA in *Sergentomyia barraudi*, *S. indica* and *S. khawi* in Thailand, suggesting that several *Trypanosoma* spp. may circulate in natural and rural habitats with domestic canines serving as reservoirs. The analysis of blood meals in engorged females provides new data about blood sources for nine sand fly species, showing opportunistic feeding habits of some, and the utilization of human blood by several, including the cavernicolous species *S. hodgsoni hodgsoni*.

## Figures and Tables

**Figure 1 insects-13-00197-f001:**
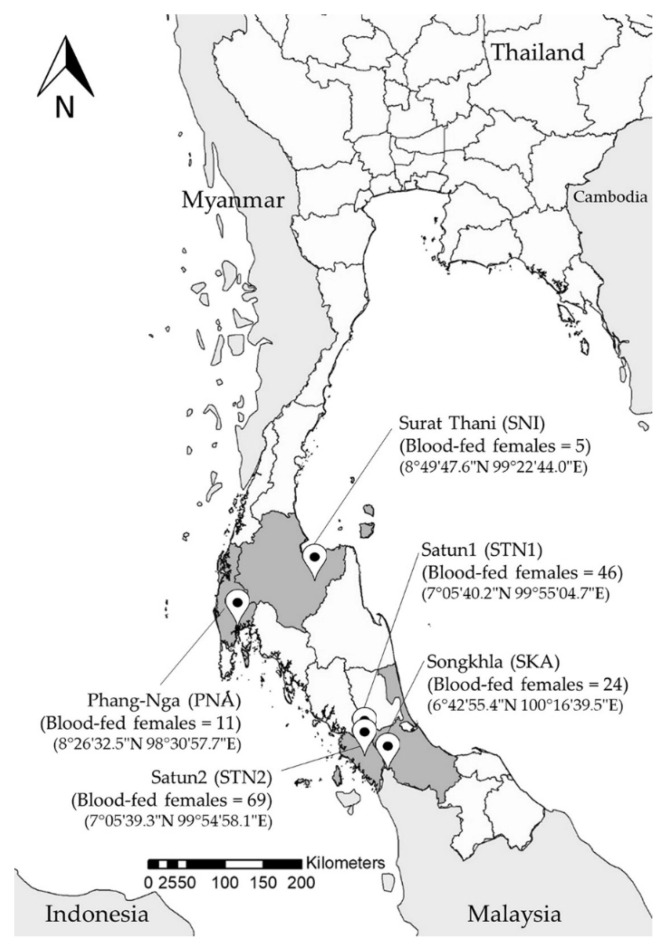
Map of sand fly sampling locations in the south of Thailand with geographic coordinates and the number of collected blood-fed female sandflies at the sites.

**Figure 2 insects-13-00197-f002:**
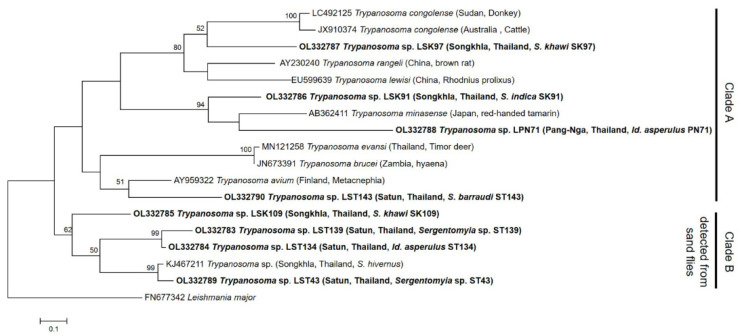
Maximum likelihood phylogenetic tree of *Trypanosoma* ITS1 region sequences. *Leishmania major* (FN677342) was used as root of tree. The number on the node represents bootstrap value (%) derived from 1000 replicates. The scale bar represents 0.1 nucleotide substitutions per site.

**Figure 3 insects-13-00197-f003:**
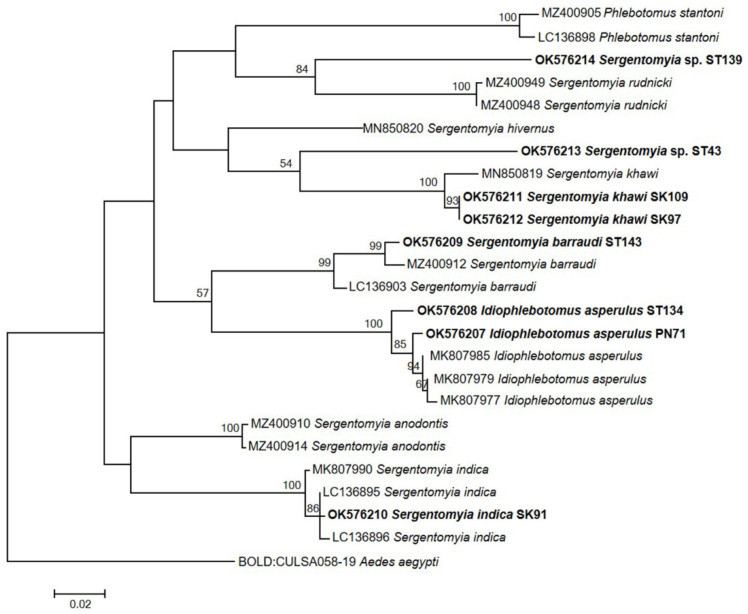
Phylogenetic tree analysis of 563 bp sand fly COI gene sequences using the maximum likelihood method. Bootstrap values above 50% derived from 1000 replication are shown on branches. The scale bar represents 0.02 nucleotide substitutions per site. Bold letters indicate samples from this study. The COI gene sequence of *Aedes aegypti* (BOLD: CULSA058-19) from the BOLD database was used as root of tree.

**Figure 4 insects-13-00197-f004:**
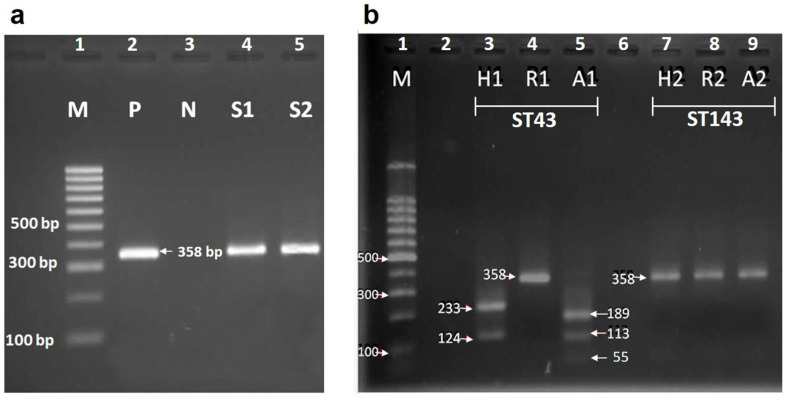
(**a**) Polymerase Chain Reaction (PCR) amplification of vertebrate DNA from *Trypanosoma* sp. infected blood-fed female sand flies using the cytochrome b (*cyt b*) gene. Lane 1: 100 bp molecular weight marker (BioLine, USA), Lane 2: positive control (genomic DNA of human blood), Lane 3: negative control (ddH_2_O), Lane 4–5: genomic DNA of blood-fed female sand flies S1 and S2 represented sample code ST43 and ST143, respectively. (**b**) Electrophoresis of partially amplified *cyt b* gene fragments (358 bp) from vertebrate host of *Trypanosoma* sp. infected blood-fed female sand flies digested with restriction enzymes *Hae* III, *Rsa* I and *Aci* I. Lane 1: 100 bp molecular weight marker (BioLine, Taunton, MA, USA), Lane 3–5: sample ST43 digested by *Hae* III, *Rsa* I and *Aci* I, respectively. Lane 7–9: sample ST143 digested by *Hae* III, *Rsa* I and *Aci* I, respectively.

**Table 1 insects-13-00197-t001:** The DNA fragments size (bp) of *cyt b* PCR products cleaved by three endonuclease enzymes.

Blood Meal Sources	Fragments Size (bp)		
	*Hae* III	*Rsa* I	*Aci* I
*Homo sapiens* (Human)	233, 124	358	189, 113, 55
*Canis familiaris* (Dog)	358	358	358
*Rattus novergicus* (Rats)	358	267, 59, 31	358
*Felis catus* (Domestic cats)	272, 74, 11	214, 119, 24	244, 113
*Sus domesticus* (Swine)	153, 130, 74	358	358
*Bus Taurus* (Cattle)	159, 124, 74	322, 31, 4	358
*Gallus gallus* (Chicken)	159, 124, 74	208, 149	308, 49
*Equus caballus* (Horse)	159, 124, 74	358	244, 113

**Table 2 insects-13-00197-t002:** Morphological identification and blood meal source analysis of blood-fed female sand flies.

Sand Fly Species	No. of Sand Flies in Collection Sites						Blood Meal Source		
	SKA	SNI	STN1	STN2	PNA	Total	Human	Dog	Rat
*Id. asperulus*	0	2	5 ^(1)^	5	7 ^(1)^	19	3	0	0
*P. argentipes*	1	0	1	1	0	3	2	0	1
*P. stantoni*	2	0	1	1	0	4	0	0	0
*S. anodontis*	0	3	8	8	1	20	2	0	1
*S. barraudi*	2	0	5 ^(2)^	2	0	9	2	1 ^(1)^	0
*S. hamidi*	0	0	19	4	0	23	7	0	0
*S. hodgsoni*	0	0	2	1	1	4	2	0	0
*S. hodgsoni hodgsoni*	0	0	0	32	0	32	4	8	11
*S. indica*	4 ^(1)^	0	0	1	0	5	0	0	0
*S. iyengari*	0	0	1	1	0	2	0	0	0
*S. khawi*	14 ^(2)^	0	2	0	1	17	2	0	0
*S. silvatica*	0	0	1	10	0	11	7	0	1
*Sergentomyia* sp.	1	0	1	3 ^(1)^	1	6	2 ^(1)^	0	0
Total	24	5	46	69	11	155	33	9	14

Collection sites near rural settlements in Songkhla province (SKA), Surat Thani province (SNI), Satun province (STN1), Pang-Nga province (PNA) and in a Buddha cave in Satun province (STN2). Superscripted numbers with parentheses represent the number of sand flies infected by *Trypanosoma* sp.

## Data Availability

All the associated data are available in the manuscript.

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
