# Peer review of "Phlebotomine Sand Flies in Southern Thailand: Entomological Survey, Identification of Blood Meals and Molecular Detection of Trypanosoma spp."

_insects, 2022, doi:10.3390/insects13020197_

Round 1

Reviewer 1 Report

Dear Editor,

Buatong et al. present a survey of sand fly species in the south of Thailand, together with an attempt to identify the blood meal source of engorged females as well as potential trypasonomatid parasites. They collected 155 engorged specimens from different rural settlements and one cave site. They performed morphological and molecular identifications of sand fly species, detection and identifications of trypanosomatid parasites using sequencing of ITS PCR amplicons, and identification of the blood meal source using PCR-RFLP in the cytB gene. They identified 12 different species of sand flies that constitute half of the species listed so far in the country, and they confirmed the carvernicolous behaviour of one of them. All sand flies appear to have fed on humans, dogs or rats (but it seems that these are the only species that can be identified with their method). They do not detect Leishmania DNA, as could be expected from sand flies. Instead, they detect what seems to be a variety of unknown Trypanosoma species.

Please find my comments below. I also attached a commented version of the manuscript with remarks and suggestions that I think should be addressed.

Major comments:

Although relatively simple, this study can contribute to the description of sand fly fauna in Thailand. The methods are mostly sound. My main criticism relates to blood meal identifications. The method they employ (PCR-RFLP) appears outdated and only target three "species" of potential hosts: humans, rats or dogs. I assume that far more potential host species can be expected from their study sites, for which they do not assess the expected RFLP patterns. Therefore, their conclusions on sand fly blood meals appear unreliable to me (especially when they conclude that even cavernicolous species appear to feed on human, cats and dogs). Why didn't they sequence the cytochrome B fragment as they did for sand flies and trypanosomatids? This would be much more interesting and I would highly encourage the authors to do so. If not possible, I would try to assess expected RFLP patterns for other potential host species (wild and domestic). Otherwise, they should at the very least to acknowledge and discuss the limitation of their approach, or even remove this part of the manuscript. In the current state, I think that these results bring little to no information, or may even be misleading and counter-productive. My second main remark relates to the diversity of unknown Trypanosoma species they seem to discover. This is highly surprising to me. Such a diversity of unknown Trypanosoma species from humans, dogs or rats is certainly unexpected (especially in Thailand where trypanosomes have been extensively studied). This raises again the question of the reliability of their blood meal identifications. In any case, this result is intriguing. It would be great if the authors could try confirming their results by sequencing another marker that can be used for Trypanosomes (this could really make their study more impacting). Otherwise, I think they should at least discuss this more extensively.

Minor comments:

See the commented pdf. Overall, I think that more references should be given along the text, and that the language could be slightly improved (especially the use of articles which are often missing).

Best regards

Author Response

Dear Reviewer #1

We really appreciate your constructive suggestions. We revised the manuscript accordingly, following your major comments.

Regarding the first comment, we have added an explanation of the potential of PCR-RFLP in the Introduction, comparing the potential of PCR-RFLP with PCR amplification + sequencing, highlighting the advantages of the PCR-RFLP technique that makes it an efficient and reliable alternative to PCR + sequencing (lower cost and quicker results). Moreover, data demonstrating the efficiency of the three restriction enzymes in this study was added in Table 1. The pattern of cleaved DNA fragments in our study identified the blood meal source from humans, rats, and dogs. We would also like to explain that human blood could be found in sand flies from a cave in Satun province, because it is a religious site that serves as monks’ residence.

Regarding the second comment, our study detected Trypanosoma DNA from engorged sand flies and identified it as Trypanosoma spp. based on the ITS1 region. For further identification at the species level, we plan to design specific primers in future work. The current report of Trypanosoma DNA detected in sand flies from southern Thailand using ITS1-PCR and SSU rRNA-PCR cannot identify the species, and novel findings are expected in the mentioned future work.

Moreover, we added more references to the manuscript where relevant, and improved the language as suggested.

Best regards,

Reviewer 2 Report

In Thailand, as mentioned by the authors, the first case of autochthonous leishmaniasis was reported 20 years ago, since then the number of human cases is increasing in different regions of the country, mostly from the southern Thailand. The authors detected  Trypanosoma sp.  DNA in three species of Sergentomyia and an Idiophlebotomus asperulus but no Leishmania spp. DNA. Furthermore bloodmeal analysis revealed Trypanosoma-positive females fed on dog and human. The current results contributes to the knowledge of leishmaniasis and trypanosomiasis transmission in southern Thailand.

The study is well-designed and organized appropriately, and the manuscript is well-written based on the results obtained. However, the following spell check  will be required: 

Line 99; replace "sans flies" by "sand flies"

Line 153; Trypanosoma, should be italic

Lines 240-253; all the specific names of the sand flies, should be italic 

Author Response

Dear Reviewer #2

         We very much appreciate your constructive suggestions for our manuscript, and we have made revisions following your comment.

Best regards,